# Human prestige psychology can promote adaptive inequality in social influence

**Thomas J. H. Morgan** [1,2,5] ✉, **Robin Watson** [1,2,3,5], **Hillary L. Lenfesty** [1,2] & **Charlotte O. Brand**[4]

Human hunter-gatherer groups were commonly thought to be broadly egalitarian, with increasingly formal hierarchical social structures hypothesized to spread following the introduction of agriculture. However, this view is being challenged by mounting evidence for social hierarchies in several foraging populations. Nonetheless, the processes by which such hierarchies emerge, and whether human hierarchies are homologous with non-human systems of dominance, remains unclear. Here we examine the role of prestige, the tendency to freely confer status and influence on skilled or esteemed individuals and a proposed component of human-unique cultural psychology, in generating unequal patterns of social influence. Through a combination of cultural evolutionary modelling, human experimentation, and evolutionary simulations, we find that human prestige psychology generates highly unequal influence hierarchies, and that the "prestige sensitivity" we measure empirically in human participants closely matches the predictions of our evolutionary simulations, suggesting it is an evolved psychological adaptation. Nonetheless, unlike non-human dominance hierarchies, the processes involved are non-coercive, being driven by individuals freely seeking high quality information. We thus conclude that social hierarchies plausibly have a deep evolutionary history in our lineage, with prestige enabling hierarchies to be mutually beneficial as opposed to coercive.

Although dominance hierarchies are common among primates[1], classic theory suggests that human hunter gatherer groups were frequently relatively egalitarian[2–5] with more complex or hierarchical social structures increasing in prevalence after the spread of agriculture[6]. The primary evidence for the egalitarian forager hypothesis comes from cross-cultural analyses that find contemporary foraging societies are more egalitarian than agricultural societies[4,7], for instance the Ju/'hoansi[8] and the Hadza[9] are both markedly egalitarian. Comparative data suggests that resource variability favors egalitarian social structures because strong food sharing norms, typical of egalitarian societies, can better manage the risks of sparse and

unpredictable food patches[10]. Additionally, egalitarian social structures in humans may be supported by stronger coalitionary aggression than that observed in other primates[11], or the greater degree of pair-bonding, kin recognition, and ties between groups[12].

Despite this evidence, the view that unequal or hierarchical populations were infrequent in human evolutionary history faces challenges[3,10]. There are several examples of populations that are (or were) not strictly egalitarian[13] and these forms of leadership may have emerged early in hominin evolution[14]. This includes archeological evidence from the Pacific Northwest coast showing signs of inequality, such as slavery and elaborate burials[15]. Leadership is also pronounced

[1]Institute of Human Origins, Arizona State University, Tempe, AZ, USA. [2]School of Human Evolution and Social Change, Arizona State University, Tempe, AZ, USA. [3]School of Psychology, Sport Science & Wellbeing, University of Lincoln, Lincoln, UK. [4]Human Behaviour and Cultural Evolution Group, College of Life and Environmental Sciences, University of Exeter, Penryn, UK. [5]These authors contributed equally: Thomas J. H. Morgan, Robin Watson. ✉ e-mail: thomas.j.h.morgan@asu.edu

in some foraging contexts, for example Inuit whale hunting[16]. Even amongst relatively egalitarian populations there is evidence of inequality in social influence[14] and reproduction[17,18]. Such inequality may bring group-level benefits[14], for instance electing powerful leaders may be a solution to the collective action problems produced by permanent settlements[19]. Other benefits may include the defense of clumped, predictable resources which are frequently associated with unequal social structures[20].

While the above shows that ecological factors are currently associated with social structure, such top-down approaches are limited by the fact that contemporary foraging groups may be unlikely to be representative of historical foraging populations as they inhabit extreme environments (though this is debated[21]) and also have contact with other agricultural societies[3]. Aside from ecological factors, individual psychological processes may also shape group structures[22] and such processes have been implicated in social inequality. For example, theoretical models have demonstrated that payoff-biased transmission can foster strong social stratification[23], while gossip can help sustain an egalitarian social structure comprised of anti-dominance coalitions[24]. Another possible mechanism is prestige, the uniquely human tendency to freely confer status and influence on skilled or esteemed individuals[25–27].

Prestigious individuals receive many benefits. For example, prestigious Tsimané men had more children, had more extra-marital affairs, married earlier to more attractive wives who had children earlier, had more allies, and more frequently got their way in group disputes[28]. Similar patterns are observed elsewhere, with high status, and in particular hunting skill, associated with preferential access to resources, increased fertility, access to additional mates, and greater political influence[17,25,29–31]. Prestigious individuals are also more influential, with the effect of prestige on social influence supported empirically, both by field data documenting cultural "big men"[32–34] and laboratory experiments[35–38]. Despite this, the impact of prestige on social structure is unclear. Nonetheless, because prestige shapes individual relationships, it plausibly affects social structures too. Moreover, because accruing followers enhances prestige, it may be subject to unusual positive feedback dynamics that generate inequality. Prestige may also explain a key difference between human and non-human hierarchies. In primate hierarchies, social structure is driven by physical aggression and the coercion of the weak by the strong[39–41]. In such a system, high ranking individuals gain fitness benefits denied to others[42–45]. By contrast, while there is cross-cultural evidence that human leadership (including in foraging groups) can be coercive[46], human societies often exhibit generous and benevolent leaders who are voluntarily deferred to by their followers[34,47,48]. Prestige, often characterized as a prosocial alternative to dominance[25,27], is hypothesized to be the source of these leaders' status[33]. A final reason to study prestige is its deep evolutionary importance to our lineage. Prestige is argued to have coevolved with the capacity for culture as well as our species' cooperative foraging and reproductive strategies, implying it is a feature of deep human history and so may have shaped hominin societies for millions of years[25,49].

Here, we make three contributions to understanding the relationship between human prestige psychology and social hierarchies (see "Methods" for details of all three). First, we use an individual-based cultural-evolutionary model that assumes the existence of prestige psychology to examine its impact on the distribution of social influence, finding that prestige psychology alone can generate groups that range from egalitarian to autocratic depending on its intensity. Second, we experimentally measure this intensity in humans, finding it to be strong enough to produce marked influence hierarchies in our model, where a small number of individuals lead decision-making. Finally, we conduct an evolutionary simulation to assess whether prestige psychology, including the intensity we measured in humans, is adaptive and so evolutionarily plausible. The results of this simulation support our experimental findings. We conclude that human prestige psychology adaptively predisposes human societies to autocratic yet non-coercive social hierarchies. This challenges the hypothesis that prehistoric human societies were egalitarian by providing a species-wide mechanism through which influence hierarchies can form.

## Results
### The cultural dynamics of prestige and inequality
We constructed an individual-based model where individuals begin with a fitness-irrelevant belief and, at each timestep, either innovate a novel belief with probability $q$, or defer to an individual and copy their belief. When individuals are deferred to by others, they linearly accrue prestige ($P$), which decays non-linearly over time at rate $p$. To examine the interaction of prestige with other factors, and to relax the assumption of a panmictic population[50], we assigned individuals to fixed, random locations in a two-dimensional space (edgeless to avoid boundary effects), with the distance between individuals reducing social influence (for visualizations see SI Fig. S1). This space can be considered as a literal surface or map, but it may also be considered a proxy for any other factors affecting influence, for instance, kinship.

On each timestep, when choosing who to defer to, the weight observer $i$ places on individual $j$ is:

$$w_{i,j} = \left(1+P_j\right)^{s_p} e^{-dD_{i,j}} \tag{1}$$

where $P_j$ is the individual's prestige, $D_{i,j}$ is the distance between the observer and individual, and $d$ is the distance penalty. When $d$ is 0, space ceases to matter and the only factor that determines whether an individual is copied is their prestige. Prestige sensitivity ($s_p$) determines how observers take account of prestige. When it is 0, observers are entirely insensitive to prestige and weight solely according to distance; when $0 < s_p < 1$ prestige affects weight, but with diminishing returns; when $s_p = 1$ weight increases in proportion to prestige; and when $s_p > 1$ prestige results in accelerating growth in weight. The distance penalty means that individuals further away are assigned less weight, but weight never reaches 0. Individuals can select themselves, but this does not contribute to their prestige.

## Table 1 | Model parameters

| Parameter | Type | Values |
|---|---|---|
| $q$, probability of innovation | Varied across model repeats | 0.001, 0.01, 0.1 |
| $s_p$, prestige sensitivity | Varied across model repeats | 0, 0.5, 1, 1.5, 2, 2.5, 3 |
| $d$, distance penalty | Varied across model repeats | 0, 1.5, 3, 5 |
| $p$, prestige decay rate | Varied across model repeats | 0.2, 0.5, 1 |
| Generations | Fixed | 400 |
| Population size | Fixed | 400 |

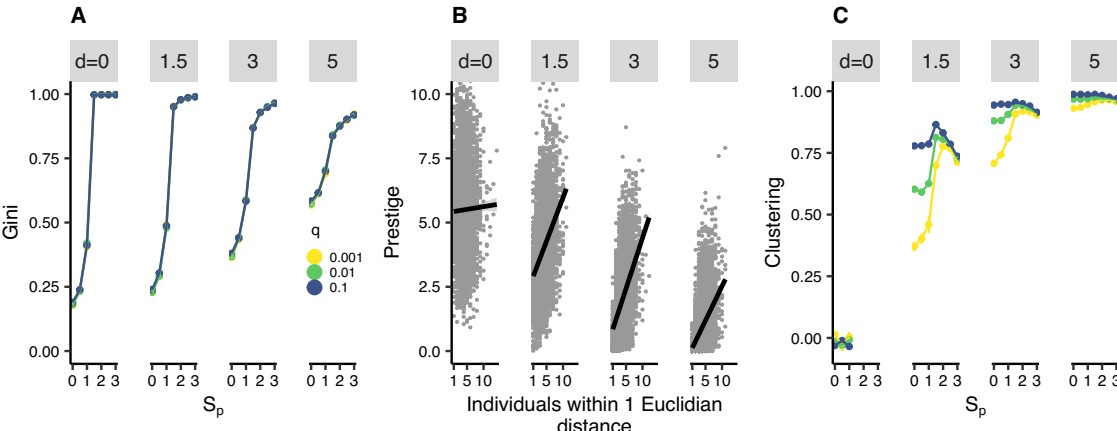

**Fig. 1 | Cultural evolutionary dynamics of prestige for different values of prestige sensitivity ($s_p$), distance penalty ($d$), and innovation rate ($q$). A** Mean and standard error of the Gini coefficient of prestige across 12 repeats of the simulation. **B** Spearman's correlations between the number of individuals within 1 Euclidean distance and an individual's prestige. Lines show the linear regression.

Correlation coefficients are: 0.023, 0.365, 0.502, and 0.501. **C** The clustering statistic (see methods for details). Less innovation decreases clustering because beliefs occasionally spread between groups, creating oddly shaped super-groups. Innovation re-fragments these into more coherent local clusters. Values for $s_p > 1$ when $d = 0$ cannot be calculated because the population is entirely homogenous.

Across model repeats, we varied: the probability of innovation ($q$), prestige sensitivity ($s_p$), the distance penalty ($d$), and the prestige decay rate ($p$) (Table 1). Note that, within each repeat, prestige sensitivity was constant and could not evolve. As such, this model explores the cultural evolutionary consequences of an assumed psychology and so does not include factors hypothesized to be the evolutionary foundation of prestige, such as individual variation in skill or ability. Later, we extend this model and explore the evolution of prestige. For every parameter combination, we repeated the model 12 times. The model reached equilibrium in under 100 generations, and so out of caution we ran all repeats for 400 generations. At the end of each repeat, social inequality was quantified using the Gini coefficient of the prestige of all individuals. We additionally assessed the spatial and temporal structuring of beliefs to assess cultural group formation. Below we only report results where $p = 0.2$, as $p$ did not affect the results. For qualitatively unchanged results exploring different population sizes and spatial configurations, see SI Figs. S5–S8.

Social inequality, as measured by the Gini coefficient, is strongly affected by prestige sensitivity, $s_p$: when it is low, populations are relatively egalitarian; when high, unequal groups emerge (Fig. 1A). This transition to inequality is moderated by the distance penalty. When $d$ is low, there is a sharp transition from egalitarian to highly unequal influence patterns when $s_p > 1.25$. In fact, without distance moderating the influence of prestige, the entire population defers to a single individual and becomes a single cultural group when $s_p \geq 1.5$ (SI Fig. S2, S3). As $d$ increases this pattern is softened; with low $s_p$, groups become somewhat less egalitarian as $d$ increases (because $d$ boosts the influence of individuals who happen to be close to many others; Fig. 1B), but with high $s_p$, groups are a little less unequal as $d$ increases (because $d$ limits the ability of prestigious individuals to exert influence across the entire population). Thus, while prestige sensitivity is the primary determinant of the (in)equality of the distribution of social influence, distance (which is a proxy for other systemic factors) somewhat moderates this and can affect the influence of specific individuals. When both $s_p$ and $d$ are low (<1.5), coherent cultural groups do not form (SI Fig. S2, S3), and the population remains egalitarian with drift being the dominant factor. Despite these effects of $d$, the effect of $s_p$ is much more powerful; to illustrate, the relationship between the Gini coefficient and $s_p$ has an $R^2$ of 0.58, whereas for $d$ it is 0.01. There is no effect of innovation rate, $q$, or the prestige decay rate, $p$ on the Gini coefficient (SI Fig. S4).

## Measuring human prestige sensitivity

Given the importance placed on prestige sensitivity by the above model, we conducted an online experiment to measure it in human participants using the software platform Dallinger (v5.1.0, https://dallinger.readthedocs.io). Eight hundred participants were recruited from MTurk and arranged into 80 groups of 10 (no statistical method was used to predetermine sample size), with participants allocated to groups in the order they arrived. Within each group participants simultaneously completed an online task where they were shown 80 blue and yellow dots and were asked to judge which was the majority color. Participants completed 50 trials. In the first 10 trials, participants made a single judgment on their own. For the remaining 40, participants made their own decision and then chose a group member to defer to and copy. To make an informed choice, participants were shown an anonymized table of their group members, containing: (1) the number of times they had been copied across all previous trials (their prestige), and (2) their accuracy over the last $N$ trials (where $N$ is the "richness" of the accuracy information). The richness of the accuracy information ($N$) varied across experimental conditions, from poor (1), to moderate (3), rich (5), and very rich (10). Each group of 10 participants was assigned in advance to one condition, and all conditions had 20 assigned groups. Accuracy information was based on their performance without social information, thus serving as a measure of their skill at the task and not their ability to copy strategically. Individuals were paid a bonus payment proportional to their own accuracy as well as the accuracy of the group members they deferred to.

We fit two Bayesian models to our data using MCMC methods in R (v4.4.3), using the package rjags (v4-17) to interface with JAGS (v4.3.2). The first was a categorical model predicting which group member participants deferred to ($N = 20,035$ decisions, with 5702, 4483, 5220, and 4630 from the poor, moderate, rich and very rich conditions, respectively). The probability of each group member being chosen was a function of a baseline, their prestige and their accuracy. Parameters determined the overall influence of prestige and accuracy in each of the four conditions ($\beta_{P,1:4}$ and $\beta_{A,1:4}$, respectively), as well as the observer's sensitivity to differences in prestige and accuracy ($s_P$ and $s_A$, respectively). The prestige sensitivity parameter, $s_P$, is directly equivalent to $s_p$ in the theoretical model. The second model was a gamma model predicting each participant's prestige based on their accuracy and the experimental condition ($N = 452$). Parameter estimates are presented as the posterior median and 95% central credible interval.

**Table 2 | Parameter estimates (median and 95% central credible interval) from the categorical analysis**

| Parameter | Interpretation | Condition | | | |
|---|---|---|---|---|---|
| | | Poor | Moderate | Rich | Very rich |
| $\beta_P$ | Weight of prestige relative to randomness | 24.6 [17.6, 34.3] | 8.5 [6.0, 12.3] | 14.7 [10.8, 19.7] | 12.3 [8.9, 16.7] |
| $\beta_A$ | Weight of accuracy relative to randomness | 7.4 [5.0, 10.7] | 12.1 [8.5, 17.4] | 35.6 [26.4, 47.5] | 30.9 [22.7, 41.8] |
| $\frac{\beta_P}{\beta_A}$ | Weight of prestige relative to accuracy | 3.35 [2.77, 4.00] | 0.70 [0.63, 0.79] | 0.41 [0.37, 0.45] | 0.40 [0.35, 0.44] |
| $s_P$ | Prestige sensitivity | 2.76 [2.61, 2.91] | | | |
| $s_A$ | Accuracy sensitivity | 22.35 [20.86, 24.12] | | | |
| $\sigma$ | Standard deviation of individual variation | 1.66 [1.56, 1.77] | | | |

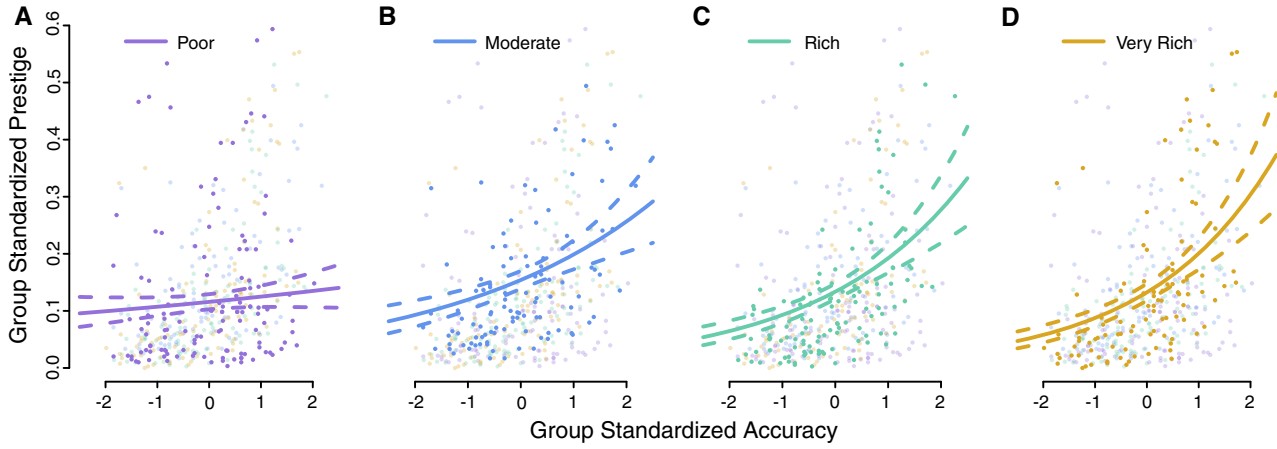

**Fig. 2 | The relationship between asocial accuracy and prestige across the four conditions. A** The poor condition. **B** Moderate. **C** Rich. **D** Very rich. Dots are individual participants, colored by condition, with data from the focal condition emphasized. Lines depict model estimates (median and 95% central credible intervals). The relationship is clearly positive in all conditions other than the poor condition.

Across all four conditions, the combined importance of prestige and accuracy was much greater than the baseline value ($\beta_{P,1:4} + \beta_{A,1:4} \gg 1$, Table 2), indicating that participants' choices of who to defer to were highly non-random. The estimate of prestige sensitivity was high ($s_P = 2.76$, [2.61, 2.91]), indicating that high-prestige participants received a disproportionate amount of the influence stemming from prestige. The same value in the above model produced highly unequal social structures. Indeed, the distribution of prestige within participant groups was highly unequal with an average Gini coefficient of 0.47 (standard error 0.02). Breaking this statistic down by condition suggests that the inequality of the prestige distribution is not affected by information richness, although the relatively few groups per condition increase uncertainty (mean Gini coefficients and standard errors are as follows: poor: 0.51, 0.05; moderate: 0.39, 0.04; rich: 0.49, 0.04; very rich: 0.51, 0.04). A coefficient of 0.5 is considered high for human populations[51], moreover, because participants only completed 40 trials, prestige inequality was likely still growing. Nonetheless, this is less than the Gini coefficients produced in the above theory, reflecting the moderating influence of accuracy[39]. Participants were also highly sensitive to accuracy differences ($s_A = 22.35$, [20.86, 24.11]), far more so than to prestige differences, such that the single most accurate participant effectively accrued all possible influence stemming from accuracy (SI Fig. S9).

We also found that the weight placed on prestige relative to accuracy ($\frac{\beta_{P,1:4}}{\beta_{A,1:4}}$) decreased with the richness of the social information. From 3.35 [2.77, 4.00] when information was poor, to 0.40 [0.35, 0.44] when information was rich. That is, participants largely used accuracy as a direct cue of skill when accuracy information was rich, but when it was poor they fell back on prestige as a collectively-produced indirect cue of skill. Such an approach is adaptive provided that prestige genuinely emerges from groups of individuals creating a powerful indirect

cue (prestige) from multiple noisy direct cues (accuracy information). The informational content of prestige is supported by the second analysis which found that participant prestige predicted their past accuracy in all but the poor condition, with the magnitude of this relationship increasing with the richness of social information (poor: $\beta_{4,1} = 0.08$ [−0.02, 0.17]; very rich: $\beta_{4,4} = 0.41$ [0.30, 0.52]; Fig. 2, SI Table S1). In the poor condition, the results are nonetheless consistent with a positive relationship between accuracy and prestige, however the evidence is weaker (94% of the posterior samples fall above 0, whereas for the other conditions 100% of the posterior samples do so). Given that the weight participants put on prestige changes across conditions, one might ask why the average Gini coefficient does not. We suggest that it is because as participants put less weight on prestige, they shifted their attention to cues of accuracy. Although accuracy is not prone to positive feedback like prestige, participants were nonetheless highly sensitive to who was the most accurate (even more so than they were with prestige), thereby causing the most accurate participant within each group to be very influential and so maintaining a high Gini coefficient.

### The evolution of prestige sensitivity

To test the robustness of our empirical results, we conducted an evolutionary simulation in which prestige sensitivity, $s_P$, evolved. Specifically, we considered a population of 2000 individuals who were randomly assigned to 100 groups of 20 and completed 40 trials of a task. Individuals varied in their ability at this task, and their fitness depended on the ability of the individuals they deferred to. On each trial, they received accurate information about how many times all individuals in their group had been deferred to, as well as stochastic information about their group members' abilities. As in the experiment, this was varied across four conditions—poor, moderate, rich and

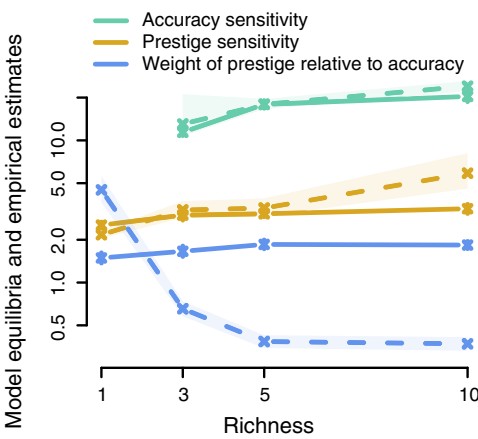

**Fig. 3 | Comparison of empirical results (dashed lines with shaded regions indicating the 95% credible interval) and predictions of the evolutionary model (solid lines with error bars indicating 2 standard errors) across information richness conditions.** The model and data are in near-perfect agreement regarding the sensitivity parameters, but diverge regarding the relative weight of prestige and accuracy. Theoretical results are the mean-of-means allele values after 5000 generations. Note the y-axis is logarithmic, for the same results with a linear y-axis see SI Fig. S10. As a rough quantification of fit, the correlation coefficients between empirical estimates and theoretical predictions across richness levels are 0.90 for prestige sensitivity, 0.95 for accuracy sensitivity, and −0.89 for the relative weight of prestige. However, as each parameter was measured at only three or four levels of richness, these statistics are underpowered and so should be interpreted with caution.

very rich—where the information was the asocial accuracy of each group member on 1, 3, 5 or 10 prior trials, respectively. Following all trials, all groups merged to form a single population and reproduced (with parents selected in proportion to their fitness) to produce a new population of 2000 individuals, and the process was repeated for 5000 generations. All individuals had a three-locus genome, with loci corresponding to their prestige sensitivity ($s_P$), accuracy sensitivity ($s_A$) and the weight put on prestige relative to accuracy (equivalent to $\frac{\beta_P}{\beta_P + \beta_A}$ in the above analysis). These loci were heritable and subject to mutation.

The results of the simulation support our empirical findings (Fig. 3, SI Fig. S10. For a sensitivity analysis see SI Figs. S11-S18). $s_P$ evolved to values around 3, while $s_A$ evolved to much higher values around 20. Unexpectedly, the simulation suggested that both $s_P$ and $s_A$ should modestly increase with information richness. To test this prediction, we reanalyzed our experimental data allowing $s_A$ and $s_P$ to vary across conditions. This additional analysis confirmed the predicted increasing pattern (SI Table S2). Unlike the experimental data, the simulations predict that individuals should always place more weight on prestige than on accuracy; $\frac{\beta_P}{\beta_A} > 1$. We suggest that participants in fact placed more weight on accuracy than prestige in the information-rich conditions because, in real populations, prestige can be distorted by irrelevant factors, something that was not possible in our simulations. Indeed, a limitation of all three elements of this work is their artificial simplicity; they treat deference as the sole source of prestige, exclude dominance-based pathways to influence, and do not incorporate other social learning biases or the richer contexts of real social groups. These constraints mean that our results capture only a subset of the processes that shape human societies, and effects of prestige psychology may differ when additional social cues, motivations, and strategic behaviors are present.

## Discussion

The classic theory suggesting that egalitarian social structures are typical of human hunter-gatherer groups is being challenged[3,10]. Here

we show that prestige—the tendency to defer to skilled or esteemed individuals[25]—can promote social hierarchies in the form of strong inequality in the distribution of social influence. In an individual-based model, we found that the sensitivity to prestige is a critical parameter, with hierarchical (or even monopolized) patterns of social influence emerging when prestige sensitivity is high (i.e., ≥2). We then experimentally measured this sensitivity among human participants, finding it to be sufficiently strong to produce such unequal prestige distributions in our model, and fostering marked influence differences in our experimental groups. Finally, we conducted an evolutionary simulation allowing prestige sensitivity to evolve. This reproduced our experimental findings and made additional predictions which were themselves confirmed by further analyses of the experimental data. Our study thus provides a comprehensive analysis of human prestige psychology and its effect on the structure of social influence, identifying a plausible psychological mechanism by which populations may develop unequal, but non-coercive, social hierarchies[3,10].

Our results complement previous work identifying simple bottom-up processes that influence social structures. For example, gossip[24], conformity[52], endorsements[53], and direct reciprocity and the tendency to assort with friends of friends[54]. In addition, prestige may also shape cooperation within groups[33] and ethnographic evidence indicates that leaders, including so-called "big men", are frequently generous, cooperative[55], and well-liked[32,34] despite also gaining status through shows of strength, warriorship and warfare[56].

We also note that the formulation of prestige in our first model is equivalent to attachment weights in models of preferential attachment processes[57]. Indeed, as both prestige and preferential attachment involve positive feedback mechanisms[25], wherein prestigious or well-connected individuals attract more prestige and connections, such similarity is intuitive and may lead to additional insights. Beyond highlighting this link, the value of our results is in quantifying how prestige sensitivity affects such positive feedback, measuring this sensitivity in humans and additionally validating it through evolutionary modeling. Nonetheless, prestige involves many mechanisms that hinder such feedback thereby differentiating it from a pure preferential attachment process. For instance, in our experiment and final model prestige was mediated by differences in the ability of individuals, which affected the eventual distribution of influence and may have softened the positive feedback cycle of prestige as did distance in the first model. There is also ethnographic evidence that social groups have established norms to keep prestige dynamics in check. For example, the Ju/'hoansi "insult" or undermine the meat brought back by hunters and credit is often shared between the hunter and the manufacturer of the arrow[58]. Similarly, among the Aché, hunters performatively mask their success by leaving kills outside the camp to be discovered later[59]. These mechanisms prevent skilled hunters from gaining excessive status. Previous theory has also highlighted that status inequalities are reduced if skilled individuals do not selectively interact with each other[50] or if status is transitive such that receiving praise from high status individuals also elevates your own status[53,60]. Data from a longitudinal study of status among forager-horticulturalists is consistent with these predictions[55], and future work could expand our models to consider cases where prestige is transitive. However, not all human status dynamics exhibit such transitivity[53], and anonymised scenarios where individuals rely on third party judgments can more easily foster disproportionate status inequalities[61]. Collectively, this suggests that prestige creates a tendency towards substantial inequality, but one that is tempered to varying degrees by social norms or other inhibitory factors depending on the specific context.

Our data also support other hypotheses regarding prestige biased transmission. For example, while prestige was strongly weighted relative to random chance, accuracy was given more weight when payoff information was rich. Although inconsistent with our

evolutionary model, this is consistent with the tendency to copy based on relative payoffs when such information is available[62,63] and the expectation that prestige is used when direct payoff information is unavailable[25,64], which has previously been documented experimentally[36]. We extend these previous results by demonstrating that this is not a discrete switch in strategy. Rather, the relative weights of prestige and accuracy steadily shift accordingly to the quality of payoff-relevant information.

Theory also supports the hypotheses that prestige is adaptive because prestigious individuals possess better than average information[25,26] and that competence is a requirement to ascend social hierarchies[49]. Nonetheless, the lack of formal theory testing the capacity of prestige cues to track skill has raised questions about the plausibility of prestige-biased transmission evolving[65]. Indeed, previous empirical evidence for this is equivocal. In online programming competitions, previously successful individuals are more likely to be copied, even if their design is not currently the best available, thus indicating an influence of their prestige[66].

However, other studies find that prestige is predictive of influence and likeability[47], but not necessarily of overall ability[67]. Our work addresses this concern in two ways. First, the genetic evolutionary model shows that prestige-biased copying can evolve, with prestige acting as a strong indirect cue of ability across a wide range of conditions (for sensitivity checks see SI, Figs. S11-S18). Second, our experiment offers an explanation for the inconsistency in prior empirical results: In all but the poor condition, we found strong evidence that prestige was a reliable indicator of a model's skill, supporting its adaptive value. However, in the poor condition, the relationship between skill and prestige was much weaker and several low-skill participants accrued significant prestige. Thus, it appears that prestige-biased transmission does tend to elevate skilled individuals, but it requires sufficient payoff-related information to do so, and when this is overly impoverished, it can fail. More generally, while prestige is likely attended to across a wide range of values of the richness of direct cues, there is likely also a middle-ground where prestige-biased transmission is particularly adaptive. When direct cues of accuracy are exceptionally rich there is little to be gained from additionally considering prestige, although prestige cues may be more readily available than direct cues of accuracy and so be attended to nonetheless. Alternatively, when accuracy cues are exceptionally poor, they cannot reliably support the collective identification of skilled individuals necessary for prestige-biased transmission to be adaptive, although prestige may remain a better cue than highly impoverished direct cues and so, again, be attended to nonetheless. Between these two extremes, prestige-biased transmission is likely highly adaptive, with prestige being a reliable indirect cue of ability far superior to direct cues.

Prestige has been argued as an alternative to dominance, with individuals ascending social hierarchies through popularity and generosity rather than coercion[25,47]. Based on our results, we suggest that despite the non-coercive nature of prestige, the great sensitivity of human prestige psychology favors hierarchical societies, much like dominance behaviors. As such, even non-coercive and prosocial psychological mechanisms like prestige can promote deeply unequal social structures. However, prestige-based hierarchies differ from the dominance-based hierarchies typical of many primates[1]. Specifically, mediated by reputational concern[68], human prestige bias can foster mutually-beneficial hierarchies in which leaders are elevated because of the value they bring to their followers. This value need not be limited to accurate information, and prestigious leaders likely exhibit a range of traits that benefit their followers, such as generosity, or effective diplomacy and leadership. On this basis, human history is not a case of egalitarian foragers being replaced by hierarchical farmers[6], but instead prestige hierarchies are part of ancient mechanisms for managing intra-group information flow and collective decision making[69]. The implicit negotiations behind such arrangements won't always produce positive outcomes; ineffective leaders may accumulate followers (as in our poor condition) and, despite contrary incentives[25], leaders may attempt to exploit followers. Furthermore, the unequal influence of individuals according to their prestige could reinforce further kinds of inequalities, including wealth or disproportionate political control. Nonetheless, the strong sensitivity to prestige we found both theoretically and empirically suggests that, for followers, the benefits of effective leadership are typically worth these risks.

## Methods
### The cultural dynamics of prestige and inequality
Consider a $20 \times 20$ toroidal space (i.e., the sides "wrap-around"), upon which 400 individuals each occupy a random fixed location. Each individual is initialized with a unique, fitness-irrelevant belief, represented as a positive integer from 1 to 400. From then on, at each timestep, all individuals update their belief: with probability $q$ they innovate and adopt a novel belief (represented as one integer higher than the previously highest belief), otherwise they defer to an individual in the population and copy their belief. Innovation occurs before copying, meaning it is possible for individuals to adopt a belief innovated by another individual in the same time step. It is also possible for individuals to select themselves, in which case they simply stick with their current belief. When individuals are deferred to, they accrue prestige, $P$, which then decays over time (though self-deference generates no prestige). The prestige of individual $i$ at timestep $T$ is:

$$P_i = \sum_{t=1}^{T} C_{i,t} e^{-p(T-t)} \quad (2)$$

where $C_{i,t}$ is the number of times they were deferred to on timestep $t$, which is weighted by an exponential decay function with decay parameter $p$ and summed across all prior timesteps ($t = [1, T]$). The decay means that, all else being equal, individuals deferred to more recently will have greater prestige than individuals deferred to less recently, and that prestige will eventually reach an equilibrium as opposed to increasing indefinitely.

When selecting an individual to defer to, an observing individual selects a target from across the entire population, with each individual weighted by their prestige, $P$, and proximity. The weight observer $i$ gives to individual $j$ is:

$$w_{i,j} = \left(1 + P_j\right)^{s_p} e^{-dD_{i,j}} \quad (3)$$

where $P_j$ is the individual's prestige, $s_p$ is the prestige sensitivity, $D_{i,j}$ is the distance between the observer and individual, and $d$ is the distance penalty. The prestige sensitivity, $s_p$ affects how observers take account of prestige. When it is 0, observers are entirely insensitive to prestige and weight solely according to distance. As $s$ increases, observers increasingly discriminate according to prestige. When $s$ exceeds 1, individuals with high prestige are disproportionately influential. The distance penalty means that individuals further away are assigned less weight, but weight never reaches 0.

Across model repeats, we varied the probability of innovation ($q$), prestige sensitivity ($s$), the distance penalty ($d$), and the prestige decay rate ($p$) (see Table 1). For every parameter combination, we repeated the model 12 times for 400 generations.

### Calculated metrics
We describe the social structures formed in the model using five metrics: (i) the Gini coefficient of prestige (a measure of inequality), (ii) the number of cultural groups (beliefs with more than 1 adherent), (iii) spatial localization (iv) the temporal stability of cultural groups,

and (v) the correlation between prestige and the number of individuals within 1 unit of Euclidian distance. For (i), (iii) and (v) see Fig. 1, for (ii) and (iv) see SI Fig. S2–S4.

The Gini coefficient (i) quantifies how prestige is distributed in the population. A value of 1 represents maximum inequality—indicating a single individual with all the prestige – while a value of 0 represents complete equality—indicating all individuals are equally prestigious.

Spatial localization (iii) quantifies whether individuals that share beliefs tend to be closer to each other than individuals that differ in their beliefs. This is summarized with a clustering statistic calculated for each individual holding each belief in the population (excluding beliefs held by only one individual):

$$\text{Clustering} = \frac{a - b}{m - b} \tag{4}$$

Where $a$ is the average Euclidean distance to those that share your belief, $b$ is the average distance to those that don't, and $m$ is the average distance to the closest $N$ individuals (where $N$ is the number of other individuals with the same belief). Clustering values approaching 1 indicate maximally spatially structured beliefs, 0 indicates no clustering, and negative values indicate "anti-clustering" meaning individuals of the same belief are unexpectedly far apart.

Temporal stability (iv) quantifies whether group membership is consistent over time. To calculate this, we identified whether each pair of individuals either shared the same belief, or did not, at generation 350 and again at generation 400, and then calculated the proportion of dyads that maintained the same degree of agreement over time.

The correlation between prestige and the number of neighbors tests the extent to which prestigious individuals are located in areas of dense population.

## Measuring human prestige sensitivity

As approved by Arizona State University IRB (Study ID: 00004815), we conducted an online experiment recruiting 800 participants through MTurk using Dallinger (https://dallinger.readthedocs.io). Recruitment was limited to participants 18 years and older, but otherwise age and sex data were not collected. Before beginning the experiment, participants were briefed, and their informed consent was collected.

Participants were placed into 80 groups of 10 and completed 50 trials, separated into an asocial task (10 trials) and a social task (40 trials). During the asocial task, participants made independent judgments. During the social task, social information from participant's group mates was available. Across four conditions, we varied the quality of this information between "poor", "moderate", "rich" and "very rich".

Participants within each group completed the experiment simultaneously. To avoid delaying groups, participants were removed if they failed to respond within 60s at any point. Of the 800 total participants, 501 (62%) completed all trials, 150 (19%) completed some trials, timed out, but completed the debriefing, and so their data is included in the analysis, and 149 (19%) either left the experiment or timed out and did not complete the debriefing, so their data was discarded.

On each trial, participants were shown an array of 80 non-overlapping yellow or blue colored dots. Their size was randomized between 10 and 20 pixels. After seeing the array for 1s, participants were asked to judge which was the majority color. To set a consistent difficulty of the task, the majority color always had 42 dots. Overall accuracy rates were around 65%. The majority color (i.e., the color with 42 dots) was randomly determined for each trial within each group.

In the 10 asocial trials, each participant made a single judgment. In the 40 social trials, after their initial judgment, participants were shown a table containing the following information about their groups' members (including themselves): (1) the number of times they had been copied across all trials (the measure of prestige), and (2) their accuracy

over the last $N$ trials. The value of $N$ controlled the richness of this information, which we varied between poor (1), moderate (3), rich (5), and very rich (10). As such, prestige information was maximally rich, while the quality of the accuracy information varied. The accuracy rating was based on judgments made before receiving social information, meaning it indicated skill at the task rather than the ability to copy strategically. After viewing the table, participants chose a group member to copy, with the chosen model's initial judgment becoming the observer's final judgment. Individuals were forced to copy on each trial, because our interest was in how individuals copied based on the information about their group mates, rather than whether they would copy at all. The social information was anonymized, and the ordering of the rows was randomized on each trial.

Participants were paid $3 for completing the experiment, plus a bonus payment for each correct judgment made (including both initial and final judgments), totaling $3 if all trials were answered correctly.

## Analysis

The data were subject to Bayesian analysis, using MCMC methods in JAGS to generate samples from posterior distributions. In all cases, a minimum of 3000 effective samples were generated from three chains, while convergence was confirmed with the Gelman-Rubin diagnostic (upper C.I. $\leq 1.01$).

We first modeled copying choices as a categorical variable. After excluding decisions from participants that did not complete the debriefing, and decisions where fewer than five participants remained in the group, 20,035 decisions, made by 638 participants, were retained. Of these decisions, 82% came from participants who completed all 40 social trials, and 94% from participants who completed at least 20 social trials.

To determine each group member's probability of being deferred to ($p_i$), they are first given an equal share of a baseline weight ($\frac{1}{D}$, where $D$ is the number of people in the group on that trial), which, in the absence of any additional factors, means group members are copied at random. Effects of prestige ($\beta_{P,C} \frac{P_i^{s_P}}{\sum P_{1:D}^{s_P}}$, where $\beta_{P,C}$ is the total weight of prestige in condition $C$, $P_i$ is the prestige of the $i$th group member and $s_P$ is the prestige sensitivity) and accuracy ($\beta_{A,C} \frac{A_i^{s_A}}{\sum A_{1:D}^{s_A}}$, where $\beta_{A,C}$ is the total weight of accuracy in condition $C$, $A_i$ is the accuracy of the $i$th group member and $s_A$ is the accuracy sensitivity) are then added on top of this. The sensitivity parameters indicate how evenly the influence stemming from prestige and accuracy was distributed among group members. In particular, $s_P$ corresponds to the prestige sensitivity parameter in the individual based model. To account for repeated measures from each participant, we include a participant-level effect $\varepsilon_Q$ that scales the effect of prestige and accuracy relative to random behavior. When $\varepsilon_Q$ is close to 0, group members behave randomly. Values of $\varepsilon_Q$ close to 1 indicate typical sensitivity to prestige and accuracy, while values above 1 indicate higher than average weighting of prestige and accuracy and so increasingly predictable behavior. Our model estimates the weights ($\beta$) and the sensitivity ($s$) parameters.

The random baseline ($\frac{1}{D}$), measure of prestige ($\frac{P_i^{s_P}}{\sum P_{1:D}^{s_P}}$) and measure of accuracy ($\frac{A_i^{s_A}}{\sum A_{1:D}^{s_A}}$) were normalized such that, across all group members, they summed to 1 (aside from the first social trial where participants were given a normalized prestige of 0 as no participants had been copied yet). The normalization served two purposes. First, it avoids the assumption that prestige (which accumulated indefinitely) becomes increasingly important relative to accuracy (which was capped at $N$) over trials. Second, it allowed the estimated weights given to prestige and accuracy to provide intuitive values of the importance of these factors relative to each other and to the baseline propensity for

random selection. For instance, if the model estimates that, in a given condition, the weight given to prestige is 0.5, and the weight given to accuracy is 2, this implies that prestige was half as influential as random chance, while accuracy was twice as influential as random chance and four times as influential as prestige.

Mathematically, the model structure is as follows:

$$\text{Group member copied} \sim \text{Categorical}(p_{1:D}) \qquad (5)$$

$$p_i = \frac{x_i}{\sum x_{1:D}} \qquad (6)$$

$$x_i = \frac{1}{D} + \varepsilon_Q\left(\beta_{P,C}\frac{P_i^{s_P}}{\sum P_{1:D}^{s_P}} + \beta_{A,C}\frac{A_i^{s_A}}{\sum A_{1:D}^{s_A}}\right) \qquad (7)$$

where $p_{1:D}$ the probability of selecting each group member. Of the recorded variables (given Roman letters), $D$ is the number of demonstrators available, $A$ is the accuracy of the group members, $P$ is the prestige of the group members, $C$ is the condition (1=poor, 4=very rich), and $Q$ is the numeric ID of the focal participant.

The priors, selected as to be weakly regularizing, are as follows:

$$\beta_{P,1:4} + \beta_{A,1:4} \sim \text{Exponential}(0.5) \qquad (8)$$

$$\frac{\beta_{P,1:4}}{\beta_{P,1:4} + \beta_{A,1:4}} \sim \text{Beta}(1,1) \qquad (9)$$

$$s_P \sim \text{Exponential}(0.5) \qquad (10)$$

$$s_A \sim \text{Exponential}(0.5) \qquad (11)$$

$$\varepsilon_{1:638} \sim \text{Gamma}\left(\frac{1}{\sigma^2}, \frac{1}{\sigma^2}\right) \qquad (12)$$

$$\sigma \sim \text{Exponential}(3) \qquad (13)$$

Note that, rather than priors being provided for $\beta_{P,1:4}$ and $\beta_{A,1:4}$ directly, they are instead provided for their sum, $\beta_{P,1:4} + \beta_{A,1:4}$, and relative proportion, $\frac{\beta_{P,1:4}}{\beta_{P,1:4} + \beta_{A,1:4}}$. This was done because participant behavior was highly non-random (i.e., $\beta_{A:P,1:4} \gg 1$) and so stating the priors this way greatly improved the sampling efficiency of the MCMC algorithm without changing results. The participant effects were modeled hierarchically, with a gamma prior assumed to have a mean of 1 (i.e., the group level parameters described the behavior of the mean participant) while the standard deviation among participants, $\sigma$, was estimated by the model. Lastly, we note that priors used do not permit negative values for $\beta_{P:A,1:4}$. As such, participants are assumed not to be repulsed by accuracy or prestige.

Although not the primary focus of this project, relevant theory predicts that prestigious individuals should be copied because prestige is a reliable indicator of accuracy[25]. To test this hypothesis, we additionally modeled the prestige on the final trial of each participant as a gamma-distributed variable that was a function of their accuracy across all fifty trials. Participants were included in this analysis only if they, and at least 5 participants in their group, completed all 40 social trials. This left a total of 452 participants in the analysis. Because group size varied, so did the opportunity to be copied and accumulate prestige. To account for this, each participant's prestige was normalized by being divided by the sum of the prestige of all participants who completed all 40 social trials in their group. In addition, because one participant was never copied (so their prestige was 0) we added

0.00001 to all prestige values as gamma distributions cannot produce 0 s. Finally, because participants were limited to copying from their group mates, we standardized accuracy scores within each group by subtracting the mean accuracy of their group and dividing by the standard deviation. As each participant contributes only a single measurement, participant-level effects are not included.

The model structure is as follows:

$$\text{Normalized final trial Prestige} \sim \text{Gamma}\left(\frac{\mu^2}{\sigma^2}, \frac{\mu}{\sigma^2}\right) \qquad (14)$$

$$\log(\mu) = \beta_{3,C} + \beta_{4,C}A \qquad (15)$$

where $\mu$ and $\sigma$ are the mean and standard deviation respectively. Of the recorded variables, $A$ is the standardized accuracy of each participant, and $C$ is the condition (1=poor, 4=very rich). Of the parameters to be estimated, $\beta_{3,C}$ is the expected prestige of a participant of group-typical accuracy, while $\beta_{4,C}$ is the log-scale effect of accuracy on prestige.

The priors, selected as to be weakly regularizing, were as follows:

$$\beta_{3,1:4} \sim \text{Normal}(0,5) \qquad (16)$$

$$\beta_{4,1:4} \sim \text{Normal}(0,2) \qquad (17)$$

$$\sigma \sim \text{Exponential}(0.5) \qquad (18)$$

As a sensitivity check, both models were run again with more diffuse priors and produced qualitatively unchanged results, thus our findings are robust to changes in the priors (SI, Figs. S19, S20 and Tables S3, S4).

## The evolution of prestige sensitivity

We consider a population of 2000 individuals who are randomly allocated to 100 groups of size 20. Within each group, individuals collectively complete 40 trials of a task. All individuals have their own constant ability to solve each trial, with these values drawn from a Beta distribution with $\alpha = \beta = 4$ at the start of each generation of the simulation. Thus, the typical individual has a 0.5 probability of solving each trial, but individuals vary and the 95% range is from 0.2 to 0.8. Individuals also have a prestige score, which is initialized at 1.

On each trial, individuals first generate a signal of their ability at the task, which is a sample from a binomial distribution with a probability of success equal to their ability, and the number of trials being the richness of social information. The latter being varied across simulations as 1, 3, 5, or 10 (thereby matching the experimental design).

Individuals use the prestige and ability signal of their group members to decide who to defer to, according to three evolving parameters: $\frac{\beta_P}{\beta_P + \beta_A}$, $s_P$ and $s_A$ which are the proportional weight of prestige relative to accuracy, prestige sensitivity and accuracy sensitivity, respectively. The probability of deferring to group member $i$ is:

$$p_i = \frac{x_i}{\sum x_{1:20}} \qquad (19)$$

$$x_i = \frac{\beta_P}{\beta_P + \beta_A}\frac{P_i^{s_P}}{\sum P_{1:20}^{s_P}} + \left(1 - \frac{\beta_P}{\beta_P + \beta_A}\right)\frac{A_i^{s_A}}{\sum A_{1:20}^{s_A}} \qquad (20)$$

Where $P$ is prestige and $A$ is the accuracy signal. A small value (0.01) is added to all accuracy signals to account for the rare situation where all accuracy signals are 0 causing $\frac{A_i^{s_A}}{\sum A_{1:20}^{s_A}}$ to be unidentifiable. This

equation matches that from the analysis of the experimental data, however the random baseline ($\frac{1}{D}$ in the experimental analysis) is removed. As such $\frac{\beta_P}{\beta_P + \beta_A}$ is the influence of prestige as a proportion of the total influence of both prestige and accuracy, implying $1 - \frac{\beta_P}{\beta_P + \beta_A}$ is the corresponding proportional influence of accuracy and meaning that no separate $\frac{\beta_A}{\beta_P + \beta_A}$ parameter is necessary. $\frac{\beta_P}{\beta_P + \beta_A}$ can readily be used to calculate $\frac{\beta_P}{\beta_A}$ as displayed in Fig. 3.

Once all individuals have chosen who to defer to on a given trial, prestige and fitness are recalculated. Prestige is increased by 1 for each individual who defers to you, and all individuals' fitness is increased by the true accuracy of the individual they deferred to. Thus, selection favors strategies that are adept at identifying high-ability group members, conditional on the past choices of other group members. An individual's final fitness, after all 40 trials, is the sum of the abilities of who they deferred to across all 40 trials, minus the sum of their sensitivity parameters ($s_P$ and $s_A$) multiplied by 0.001. There is thus a small fitness cost associated with prestige and accuracy sensitivity. This fitness cost was deliberately small and was introduced because otherwise the $s_A$ parameter evolved to very high values and then (because once a sensitivity parameter is high, large absolute differences are unimportant) was subject to strong drift. By imposing a small cost, this drift was reduced while the sensitivity parameters still evolved to similar values.

After final fitness is calculated, individuals reproduce, replacing the entire population with a new generation of 2000 offspring. Reproduction was sexual. For each offspring, two parents were selected by sampling from the parental generation weighted by fitness. A random value was then drawn from a uniform distribution ranging between 0 and 1, and the offspring allele values were set to those of the first parent multiplied by this value, plus those of the other parent multiplied by one minus this value. In this way, the average offspring was an equal blend of both parents, but offspring typically resembled one parent more than the other. Note that offspring do not inherit their task ability from their parents. Inheritance was also subject to mutation, such that inherited offspring allele values were changed by the addition of a value drawn from a normal distribution with mean 0 and standard deviation 0.05. As such, offspring could have allele values greater than or lower than those of both their parents. The mutation rate was chosen to allow the simulations to reach equilibrium within a reasonable timeframe and without introducing excessive drift. In addition, because small differences in the sensitivity parameters matter less as the sensitivity parameters increase in value, they mutated on the $\log_{10}$ scale, such that:

$$s'_P = 10^{\log_{10}(s_P) + \mathcal{N}(0,\, 0.05)} \tag{21}$$

Finally, once all offspring were created, their abilities at the task were randomly generated, their prestige was set to 1, their fitness set to 0, and the process repeated for a total of 5000 generations. Sixteen simulations were executed for each of the four social information richness values.

### Reporting summary
Further information on research design is available in the Nature Portfolio Reporting Summary linked to this article.

## Data availability
All raw, anonymized data and analysis files are available at https://doi.org/10.17605/OSF.IO/7YRTS.

## Code availability
All model and analysis code is available at https://doi.org/10.17605/OSF.IO/7YRTS.

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

## Acknowledgements

This work was funded by DARPA cooperative agreement D17AC00004 and Templeton World Charity Foundation award 20648 to T.M. We thank Rob Boyd, Gillian Hadfield and members of the BCDs discussion group for feedback.

## Author contributions

R.W., H.L. and T.M. designed and executed the cultural evolutionary model; H.L., C.B. and T.M. designed and created the experiment; T.M. executed the experiment, analyzed the data, and created and executed the evolutionary model; R.W. and T.M. wrote the manuscript; all authors provided feedback on the manuscript.

## Competing interests

The authors declare no competing interests.
