## [Transparent Peer Review file · Nature Communications]

Human Prestige Psychology Creates Adaptive Inequality in Social Influence

Corresponding Author: Dr Thomas Morgan

Version 0:

Reviewer comments:

Reviewer #1

(Remarks to the Author)

The manuscript combines formal modelling with a laboratory experiment to quantify the relative influence of conformity and payoff-biased imitation on reputation-building. Estimating both weights within the same experimental design (Table 2) is an ambitious and valuable step beyond simply demonstrating that each force matters. The authors further reinforce their empirical findings by showing that a theoretical evolutionary model reproduces the main qualitative patterns in the data. Taken together, the work has the potential to make a solid contribution to the literature on social learning and the emergence of prestige-based inequality.

That said, several issues should be addressed before the paper is suitable for publication.

- The title is confusing and needs to be changed. The model produces popularity and expertise gradients, not coercive “hierarchical societies.” A more accurate title might refer explicitly to “prestige-based inequality” or “reputation hierarchies” rather than invoking broad social hierarchies created “without coercion,” which the study does not analyze.
- The space component of the model in which individuals are frozen at their positions is completely unrealistic and unnecessary. The agents location plays no role outside Part 1, yet it adds assumptions and notation. Because Parts 2–3 succeed without this component, consider eliminating this component completely.
- The copying rule used in part 1 is mathematically equivalent to a preferential-attachment process in networks theory; heavy-tailed popularity is therefore an expected outcome. The manuscript should discuss this connection -- especially how the added expertise dimension modifies classic preferential-attachment predictions -- and clarify what is novel.
- Because skill recognition becomes crucial later, either (a) extend Part 1 so that expertise influences copying there as well, or (b) explain why it is deliberately omitted and why that omission does not undermine subsequent comparisons.
- The logarithmic scale in Figure 3 compresses visual differences between theory and data. Consider adding a linear-scale inset or supplying fit statistics to let readers judge the quality of the match.
- The match between empirical estimates and modeling predictions for the accuracy parameters is impressive indeed. Have the authors explored the sensitivity of their theoretical results to changes in parameters and distributions?

Reviewer #2

(Remarks to the Author)

The authors present a multi-faceted study (dynamic model, online experiment, evolutionary simulation) of the relationship between prestige-bias and hierarchical structure of groups. The rarity of tests of this relationship, and the comparison across model and experimental results make this paper valuable. I have no major critiques of the study design(s) but have a few critiques of the framing of the study’s motivations and interpretation of results. These include description of the absence/presence of hierarchical structures in traditional human societies (past and present), and relationship to existing work on prestige and hierarchical structure which explore the conditions under which hierarchy will be more or less

exacerbated by prestige dynamics. See below.

Lines 19-20: This implies most (anthropologists?) think formalized hierarchical social structures are limited to post-agriculture. Rather, my sense of the predominant view is that the frequency of formalized hierarchical social structures shot up dramatically (with some time lag) post introduction of agriculture. And probably no anthropologist would argue informal hierarchies are absent in even the most egalitarian societies (see here: <https://www.sciencedirect.com/science/article/pii/S1048984317301376>).

Lines 27-28: Rephrase for clarity: “and that its measured sensitivity closely matches the predictions of evolutionary theory”. Sensitivity to what? And which evolutionary theory?

Lines 40-42: Normative enforcement of food-sharing is an incomplete explanation. I would also emphasize coordination abilities not possessed by other primates, in particular larger more effective and longer-lasting leveling coalitions (see Boehm, Gavrilets, Tomasello, Marlowe). Other likely contributors to human egalitarianism include pair-bonding (see Chapais) which reduces mating competition and creates ties across groups facilitating the ability to “vote with one’s feet”.

Lines 44-49: The line of argument here is somewhat misleading in that we’ve known for a long time about inequality in Pacific Northwest societies. Or the gerontocracies in Australian hunter-gatherers. And other cases. What remains debated/“is being challenged” is the relative frequency of such inequalities in hunter-gatherers historically, even into the Pleistocene. The Pacific Northwest societies may be the tip of the iceberg of such societies, principally located along coasts and productive rivers, which were some of the first to be displaced by agriculture. Recent evidence is also showing that even in more “egalitarian” hunter-gatherers, there is non-trivial inequality in influence during collective decision-making (<https://www.sciencedirect.com/science/article/pii/S1048984317301376>) and in reproduction (<https://www.pnas.org/doi/10.1073/pnas.1606800113>). Non-coercive prestige dynamics may be even more relevant to explaining such hierarchy, in the absence of the wealth that characterizes more sedentary, “complex” hunter-gatherers.

Lines 55-56: Regarding “extreme environments”, there remains debate here too. See this paper for an alternative perspective: <https://www.sciencedirect.com/science/article/pii/S2352409X18307302>

Lines 56-62: I wouldn’t say an “alternative” approach in that psychological processes respond (often by design) to socioecological variation. See next comment for relevant example.

Lines 64-75: There is work I would discuss on how prestige processes relate to inequality, given social network structure or socioecology. When accrual or maintenance of prestige depends heavily on widespread cooperation in small communities (as is common in hunter-gatherers with food interdependence and strong social leveling mechanisms), then you are likely to see feedback between prestige and inequality attenuated if not checked. This is because prestige effectively diffuses via cooperation, whether due to the transfer of knowledge or resources or reputation gain from association with higher-prestige individuals. In support, models find that when homophily by prestige is low (<https://academic.oup.com/beheco/article/25/1/58/222376>) and when prestige diffuses across group members via their interactions (https://papers.ssrn.com/sol3/papers.cfm?abstract_id=1646328), the self-reinforcing nature of prestige is curtailed. The following longitudinal, empirical study of forager-horticulturalists is consistent with these models, finding evidence consistent with status diffusion through a community’s social network, via cooperation between lower and higher status individuals: <https://pubmed.ncbi.nlm.nih.gov/31387506/>. Conversely, in larger communities where homophily by prestige is higher, and where individuals have to rely heavily on third-party judgments of prestige, it’s more likely that prestige-based inequality will be self-reinforcing (i.e. Matthew effects), and that prestige differences are exacerbated relative to the actual value that individuals could provide to others (e.g. <https://www.pnas.org/doi/full/10.1073/pnas.1316836111>).

Lines 76-77: Prestige may have roots in non-human partner choice or competence evaluations (e.g. your reference #45). And in humans, greater interdependence in food production and in reproduction, with shifts to greater hunting and gathering and more human-typical life histories, presents expanded opportunities for prestige not just related to the capacity for culture. More generous food-sharers or those with ability to coordinate others well gain prestige independent of any information transfer.

Lines 92-101: So each generation, every individual assigns every other individual on the toroidal grid a weight according to this formula?

Line 95: In each generation, does prestige of j accrue linearly with number of individuals deferring to them?

Lines 131-133: The mathematical model results basically indicate that the more individuals are sensitive to prestige, the more inequality in prestige is expected. Are there reasons why this might not be straightforwardly predictable? Varying how number of individuals deferring to a given individual affects prestige gain may add more interesting dynamics. And related to my above comment about how diffusion of prestige may affect resulting inequality, it would be useful to allow individuals to gain prestige in proportion to the prestige of those they defer to (given its empirical frequency: <https://www.pnas.org/doi/abs/10.1073/pnas.2015188118>).

Lines 151-152: I’d clarify “richness” of the accuracy information as number of trials for which others’ performance are observable, and state how many trials associated with each of the four conditions.

Lines 181-186: Why individuals should rely on prestige when they can’t compare it to performance (“poor” information

quality) is a puzzle. Are participants assuming others could pick up on skill even in the context of poor information quality? Or this could be a product of participants being required to choose a group member to copy, which they otherwise wouldn't have done under conditions of poor information quality.

Lines 181-186: I'd report the Gini coefficients for prestige specific to each of the four conditions. The prediction being that prestige inequality should decline as individuals attend to accuracy information more (thus mitigating Matthew effects from prestige-bias).

Lines 240-242: Inequality in who is copied isn't the same as inequality in political decision-making. So I would make this caveat in terms of the relevance of your results to interpretation of hierarchy emergence in the absence of coercion. Also, political deference may be the price to pay for copying someone, but then why should such deference persist beyond the time required to copy the trait in question? It may be that considerable time is required for copying. Considering other determinants of prestige can be helpful here- that individuals defer to others who can be sources of benefit (as a mate, sharing partner, ally, leader) even if those benefits don't come from transferrable skills or knowledge.

Lines 251-253: Big Men may also be feared, which blurs the lines between dominance and prestige. It is common that Big Men gained influence via a reputation for warriorship, which they then translate into coordination of (competitive) gift-giving later in adulthood, e.g. see <https://pubmed.ncbi.nlm.nih.gov/30868368/>. Also, Big Man leadership is more formalized and competitive relative to leadership that emerges in more egalitarian societies. The label of "Big Man" is generally restricted to those leaders of horticultural or pastoral societies where there is more frequent coalitional violence and more wealth to contest, with greater demand for conflict resolution and collective action. Nevertheless, there is ethnographic evidence that status/influence in more egalitarian settings is also dependent on cooperation, and vice versa cooperation is dependent on the distribution of influence: <https://royalsocietypublishing.org/doi/10.1098/rspb.2019.1367>

Lines 257-263/272-276: I recently came across this paper which discusses how the adaptiveness of prestige-bias may depend on presence of other learning mechanisms (https://brill.com/view/journals/jocc/24/5/article-p466_3.xml). It's a puzzle why prestige-bias is adaptive in cases where information quality is poor, as this is when one might expect distribution of prestige to be more decoupled from the distribution of performance, as you found in your experiment! And when information quality is high, why rely on prestige bias at all? Can you address this?

Reviewer #3

(Remarks to the Author)

Review of "Human Prestige Psychology Creates Hierarchical Societies without Coercion.

Overall, this is a very interesting paper and should be published. Using both simulation and experimental data it substantiates a speculation that has existed since the Prestige-Dominance Theory was first developed in the early 2000s.

The biggest challenge the paper confronts is length. In my view, it would be ideal if the authors had more space. Everything feels rather compressed. The paper has multiple simulations and an experiment.

Given space limitations, some of suggestions for material to add might be difficult. But, I'll put it out there anyway.

On the title: the authors make the case for prestige and individual inequality (Gini). It's important to individual differences in payoffs separate from hierarchy, political power and stratification. Among "big men" societies, the "big man" is in a "first among equals" situation. There's no stratification or hereditary power, etc. Anthropology is full are great examples, but I like the Handbook of California Indians for cases. These population are all foragers, but there are clearly "rich men" and wealthy prestigious foragers.

Anyhow, you should make all this clear. I'd consider sticking to social and economic inequality at the individual level and avoid stratification or hierarchy (which is in the proposed title).

I also think readers would be helped by learning a little about how prestige is suppressed in foraging societies. Among the Ju'hoasi, the camp "insults" the meat to the hunter, as a way of "cooling his heart" (even if they are delighted). Similarly, credit is shared due to 'arrow sharing' so both the hunter and the owner of the arrow get credit for the kill. This spreads the status bump. Further, hunters who get a couple of kills in a row, take time off so has not to accumulate to much prestige. Boasters are shot-down with jokes if they get too big for their waistcoats. You can see how the norms operate to suppress the prestige that might otherwise result in inequality. This seems important for the argument.

I'd also recommend explaining the deference benefits that the prestigious get...additional wives, matings ("wife loaning"), influence at meetings, help cutting canoes (Andaman Islanders). Having many followers improves security, given endemic warfare. This is useful because the simulation and experiments are pretty abstract.

The main simulation is challenging to explain. I think you can do better. First, create an image of the toroidal grid and show your individuals on the grid. I like Mathematica for this. Some readers were be perplexed by a toroidal grid. In addition to showing it, you should say we you used this. Also, explain why you didn't create communities, like villages, camps or territorial communities (like chimps and bonobos). You seem to spread individuals across space on the donut.

Two notes: (1) running for 400 generations seems odd. Why not run it until a stable distribution is achieved?

Isn't it crucial to vary the population size? Seems like we want to know how pop size influences things. Some people think repeatedly running a small pop size like 400 is equivalent to running a larger pop size, but its not. Well, sometimes it is, but it can fool you.

I wonder if Figure 1 can expanded to deliver a more complete image of the results.

Experiments—very cool. The experiment confirms the hypothesis model in the simulation. It also nicely confirms the more fundamental prestige-dominance theory.

Please tell us who these participants are in the main text. I see the info in the supplemental (Mturk and Dallinger), but this is important info. Banishing it to the supplemental implicitly communicates to junior scholars that this represents irrelevant detail.

Version 1:

Reviewer comments:

Reviewer #1

(Remarks to the Author)

I am quite happy with the revision.

Reviewer #2

(Remarks to the Author)

Thank you for responding in detail to my comments. I have no further comments/suggestions. I remain in support of publication.

Reviewer #3

(Remarks to the Author)

The revision looks great. I recommend publication.

REVIEWER COMMENTS

Reviewer #1 (Remarks to the Author):

The manuscript combines formal modelling with a laboratory experiment to quantify the relative influence of conformity and payoff-biased imitation on reputation-building. Estimating both weights within the same experimental design (Table 2) is an ambitious and valuable step beyond simply demonstrating that each force matters. The authors further reinforce their empirical findings by showing that a theoretical evolutionary model reproduces the main qualitative patterns in the data. Taken together, the work has the potential to make a solid contribution to the literature on social learning and the emergence of prestige-based inequality.

That said, several issues should be addressed before the paper is suitable for publication.

We thank the reviewer for their positive assessment of our work and for their helpful and thorough comments which have significantly improved the manuscript. We describe how we have addressed them below, and we hope the reviewer considers our revisions satisfactory.

- The title is confusing and needs to be changed. The model produces popularity and expertise gradients, not coercive “hierarchical societies.” A more accurate title might refer explicitly to “prestige-based inequality” or “reputation hierarchies” rather than invoking broad social hierarchies created “without coercion,” which the study does not analyze.

We thank the reviewer for this suggestion and have revised the title to “Human Prestige Psychology Creates Adaptive Inequality in Social Influence”. This contains the core topic of our work (prestige and social influence), and refers to the distribution of social influence (“Inequality”), as well as our empirical component (“Human”) and the final evolutionary model (“Adaptive”). In addition, it avoids the invocation of broad social hierarchies or coercion.

Beyond the title, we have also made clear throughout the manuscript that when we use the term hierarchies it is specifically in reference to influence hierarchies (e.g., lines 25-31, 300-302).

- The space component of the model in which individuals are frozen at their positions is completely unrealistic and unnecessary. The agents location plays no role outside Part 1, yet it adds assumptions and notation. Because Parts 2–3 succeed without this component, consider eliminating this component completely.

Thank you for this comment and we sympathize with the reviewer's concern. After careful consideration we have opted to keep space in the model, nonetheless we have made revisions to make our motivations for doing so clear (lines 114-119, 160-164).

One reason for keep space in the model is that while R1 suggests it be removed, R3 asked us to keep it in and in fact expand on the spatial configurations considered. This puts us as authors in a somewhat tricky position, and we decided that keeping the current treatment of space in the main manuscript, with the additional configurations in the SI was the best compromise between the requests of R1 and R3.

In addition, there are purely scientific motivations for keeping space in the model. Primarily that without space there are no other factors systematically affecting copying decisions other than prestige. Thus, distance allows us to assess the robustness of the effect of prestige on the distribution of social influence to the presence of other factors (with the case of $d=0$, as we now make clear, corresponding to the absence of other factors, lines 126-127). We agree that this treatment of space is necessarily unrealistic, however, as we now additionally make clear, we use space not as a depiction of how individuals move in the real world, but rather as a proxy for factors other than prestige that may influence who is copied (lines 114-119).

Secondarily, the inclusion of space also generates the result (shown in Figure 1B) that particularly well-connected individuals accumulate more prestige. This indicates that while the prestige sensitivity parameter broadly determines the inequality of social influence, other factors can nonetheless shape which individuals rise to prominence. We now discuss this in more detail (lines 161-164).

- The copying rule used in part 1 is mathematically equivalent to a preferential-attachment process in networks theory; heavy-tailed popularity is therefore an expected outcome. The manuscript should discuss this connection -- especially

how the added expertise dimension modifies classic preferential-attachment predictions -- and clarify what is novel.

Thank you for highlighting this. We agree there is a clear relationship to preferential-attachment and that, as such, the mere production of inequality in social influence is not surprising (lines 320-324). We also discuss ways in which prestige differs from preferential attachment in having additional self-limiting processes (lines 326-345). Finally, we have clarified that the value of our work lies in the application of this process to prestige biased social learning, the quantification of the relationship between prestige-sensitivity and such inequality, the measurement of this in humans, and the validation of it through evolutionary theory (lines 324-326).

- Because skill recognition becomes crucial later, either (a) extend Part 1 so that expertise influences copying there as well, or (b) explain why it is deliberately omitted and why that omission does not undermine subsequent comparisons.

We thank the reviewer for highlighting this point. We have clarified that skill was deliberately omitted from part 1 (lines 136-140). In brief, the purpose of part 1 is to explore the consequences of prestige-biased social learning, rather than its *evolution*. Nonetheless, we agree with the reviewer that skill is a vital part of the evolution of prestige, which is why we include it in parts 2 and 3 where we no longer assume the presence of prestige psychology. Thus, the different parts of the study offer complementary investigations of different aspects of prestige.

- The logarithmic scale in Figure 3 compresses visual differences between theory and data. Consider adding a linear-scale inset or supplying fit statistics to let readers judge the quality of the match.

We appreciate the reviewers concern with log-scaled figures. We had originally adopted a log-scaled y-axis due to the different scales over which parameters range (accuracy sensitivity ranges between 10 and 25, the others are 6 and below). As such, a linear y-axis creates a lot of empty space while compressing prestige sensitivity and the relative weight put on prestige at the bottom of the figure. We have addressed this issue as follows: (1) We have created a two-panel linear-scale figure, with accuracy sensitivity in its own panel. The size of this figure means it cannot be placed as an inset within Figure 3, as such we have put it in the SI and refer to it from both the main manuscript (lines 274-275) and figure caption (line 291-292). (2) We have added correlation coefficients to the Figure 3 caption as fit statistics (lines 292-295). However, given the small number of richness levels considered

we make clear such statistics should be interpreted with caution. Collectively this gives the reader a firm basis to judge the quality of the match between theoretical predictions and empirical measurements.

- The match between empirical estimates and modeling predictions for the accuracy parameters is impressive indeed. Have the authors explored the sensitivity of their theoretical results to changes in parameters and distributions?

We appreciate the importance of checking the sensitivity of model results. We had not already done so for two reasons: (1) The model was intended as a test of the experimental results, and so where parameters were set we chose values corresponding to our experiment, and (2) the model is very slow to run making a sensitivity analysis logistically difficult. However, following the reviewer's request we rewrote our model to improve execution time (essentially rewriting the bottlenecks in C++ instead of R) which enabled us to explore more of parameter space. Specifically, we varied (1) population size, (2) group size, (3) the scale of inter-individual variation in ability and (4) the number of trials individuals completed. In all cases the results are minimally changed and so we are confident that the good fit between evolutionary predictions and our empirical results is a genuine result. The results of these sensitivity analyses are presented in the SI, Figures S11-18, and we also refer to them from the main manuscript (lines 274-275).

Reviewer #2 (Remarks to the Author):

The authors present a multi-faceted study (dynamic model, online experiment, evolutionary simulation) of the relationship between prestige-bias and hierarchical structure of groups. The rarity of tests of this relationship, and the comparison across model and experimental results make this paper valuable. I have no major critiques of the study design(s) but have a few critiques of the framing of the study's motivations and interpretation of results. These include description of the absence/presence of hierarchical structures in traditional human societies (past and present), and relationship to existing work on prestige and hierarchical structure which explore the conditions under which hierarchy will be more or less exacerbated by prestige dynamics. See below.

We thank the reviewer for their positive appraisal of our work and for their helpful comments. They have greatly improved the manuscript, particularly the introduction. We describe our revisions below and we hope that reviewer considers them satisfactory.

Lines 19-20: This implies most (anthropologists?) think formalized hierarchical social structures are limited to post-agriculture. Rather, my sense of the predominant view is that the frequency of formalized hierarchical social structures shot up dramatically (with some time lag) post introduction of agriculture. And probably no anthropologist would argue informal hierarchies are absent in even the most egalitarian societies (see here: <https://www.sciencedirect.com/science/article/pii/S1048984317301376>).

We thank the reviewer for raising this point. We agree and have softened the language in the abstract to better reflect this (lines 20-23). We have also softened that language in the introduction (lines 37-39) and cite the suggested paper (lines 50-52).

Lines 27-28: Rephrase for clarity: “and that its measured sensitivity closely matches the predictions of evolutionary theory”. Sensitivity to what? And which evolutionary theory?

We have clarified that it is sensitivity to prestige and our own evolutionary theory (lines 26-31).

Lines 40-42: Normative enforcement of food-sharing is an incomplete explanation. I would also emphasize coordination abilities not possessed by other primates, in particular larger more effective and longer-lasting leveling coalitions (see Boehm, Gavrillets, Tomasello, Marlowe). Other likely contributors to human egalitarianism include pair-bonding (see Chapais) which reduces mating competition and creates ties across groups facilitating the ability to “vote with one’s feet”.

Thank you for these suggestions, we have added them, along with supporting references, to the manuscript (lines 45-47).

Lines 44-49: The line of argument here is somewhat misleading in that we’ve known for a long time about inequality in Pacific Northwest societies. Or the gerontocracies in Australian hunter-gatherers. And other cases. What remains debated/”is being

challenged” is the relative frequency of such inequalities in hunter-gatherers historically, even into the Pleistocene. The Pacific Northwest societies may be the tip of the iceberg of such societies, principally located along coasts and productive rivers, which were some of the first to be displaced by agriculture. Recent evidence is also showing that even in more “egalitarian” hunter-gatherers, there is non-trivial inequality in influence during collective decision-making (<https://www.sciencedirect.com/science/article/pii/S1048984317301376>) and in reproduction (<https://www.pnas.org/doi/10.1073/pnas.1606800113>). Non-coercive prestige dynamics may be even more relevant to explaining such hierarchy, in the absence of the wealth that characterizes more sedentary, "complex" hunter-gatherers.

Thank you for these helpful clarifications. We have revised the introduction following the reviewer’s suggestions and added the references they suggested (lines 49-59).

Lines 55-56: Regarding “extreme environments”, there remains debate here too. See this paper for an alternative perspective:

<https://www.sciencedirect.com/science/article/pii/S2352409X18307302>

Thank you for highlighting this, we now state that the extremity of the environments occupied by modern foragers is debated and have added reference suggested (lines 61-64).

Lines 56-62: I wouldn’t say an “alternative” approach in that psychological processes respond (often by design) to socioecological variation. See next comment for relevant example.

We agree that this wording was unclear for the reasons the reviewer highlights and have revised our language accordingly (lines 61-64).

Lines 64-75: There is work I would discuss on how prestige processes relate to inequality, given social network structure or socioecology . When accrual or maintenance of prestige depends heavily on widespread cooperation in small communities (as is common in hunter-gatherers with food interdependence and strong social leveling mechanisms), then you are likely to see feedback between prestige and inequality attenuated if not checked. This is because prestige effectively diffuses via cooperation, whether due to the transfer of knowledge or resources or reputation gain from association with higher-prestige individuals. In

support, models find that when homophily by prestige is low (<https://academic.oup.com/beheco/article/25/1/58/222376>) and when prestige diffuses across group members via their interactions (https://papers.ssrn.com/sol3/papers.cfm?abstract_id=1646328), the self-reinforcing nature of prestige is curtailed. The following longitudinal, empirical study of forager-horticulturalists is consistent with these models, finding evidence consistent with status diffusion through a community's social network, via cooperation between lower and higher status individuals: <https://pubmed.ncbi.nlm.nih.gov/31387506/>. Conversely, in larger communities where homophily by prestige is higher, and where individuals have to rely heavily on third-party judgments of prestige, it's more likely that prestige-based inequality will be self-reinforcing (i.e. Matthew effects), and that prestige differences are exacerbated relative to the actual value that individuals could provide to others (e.g. <https://www.pnas.org/doi/full/10.1073/pnas.1316836111>).

Thank you to the reviewer for these helpful suggestions. We agree that these are important factors that moderate the runaway dynamics associated with prestige. This point was also raised by reviewer 3, who recommended that we include more discussion on prestige suppressing mechanisms in hunter gatherer populations, and also relates to review 1's point about the similarities and differences between prestige and preferential attachment. We have added a new section to the discussion (lines 326-345) based on these suggestions.

Lines 76-77: Prestige may have roots in non-human partner choice or competence evaluations (e.g. your reference #45). And in humans, greater interdependence in food production and in reproduction, with shifts to greater hunting and gathering and more human-typical life histories, presents expanded opportunities for prestige not just related to the capacity for culture. More generous food-sharers or those with ability to coordinate others well gain prestige independent of any information transfer.

We agree and have expanded this sentence accordingly (lines 90-93).

Lines 92-101: So each generation, every individual assigns every other individual on the toroidal grid a weight according to this formula?

Yes, that is correct. We have revised the manuscript to make this clear (lines 111-113, 121-126).

Line 95: In each generation, does prestige of j accrue linearly with number of individuals deferring to them?

Yes, that is correct. We have clarified this in the main text, as well as adding that the decay of prestige is non-linear (lines 113-114).

Lines 131-133: The mathematical model results basically indicate that the more individuals are sensitive to prestige, the more inequality in prestige is expected. Are there reasons why this might not be straightforwardly predictable? Varying how number of individuals deferring to a given individual affects prestige gain may add more interesting dynamics. And related to my above comment about how diffusion of prestige may affect resulting inequality, it would be useful to allow individuals to gain prestige in proportion to the prestige of those they defer to (given its empirical frequency: <https://www.pnas.org/doi/abs/10.1073/pnas.2015188118>).

We thank the reviewer for this suggestion and agree that the most basic result of model 1, that increasing prestige sensitivity increases the inequality in the distribution of influence, is intuitive. However, as we noted in response to a comment from reviewer #1, our work is not limited to this qualitative result, and its value comes from the quantification of the relationship between prestige-sensitivity and such inequality in model 1, the measurement of this sensitivity in humans through the experiment, and the validation of this value through evolutionary theory in model 2. We have now clarified this in the manuscript (lines 324-326).

We thank the reviewer for the additional reference they suggested and have incorporated it into the paper (lines 331-345) where we describe various mechanisms that prevent runaway prestige hierarchies from forming. We also appreciate the reviewer's suggestion to consider the impact of transitivity in prestige gain (i.e. being endorsed by a high-ranking individual benefits your rank more than being endorsed by a low-ranking individual). We have expanded our discussion to specifically consider transitivity and note that empirical evidence finds transitivity in some human networks, but not others (lines 336-343). As such, although we do not include transitivity in our model, we note that future work could do so (lines 338-340)

Lines 151-152: I'd clarify "richness" of the accuracy information as number of trials for which others' performance are observable, and state how many trials associated with each of the four conditions.

We have clarified richness as suggested, and made clear that each group was assigned to a condition with all conditions having the same number of groups (lines 188-194). We have also provided the precise number of trials associated with each of the four conditions (lines 199-201).

Lines 181-186: Why individuals should rely on prestige when they can't compare it to performance ("poor" information quality) is a puzzle. Are participants assuming others could pick up on skill even in the context of poor information quality? Or this could be a product of participants being required to choose a group member to copy, which they otherwise wouldn't have done under conditions of poor information quality.

We have added to this section of the manuscript to clarify this result (lines 374-383). Specifically, we believe that individuals in the poor condition are attending to prestige because it is constructed following multiple noisy cues of accuracy (on each trial, all group mates receive such noisy cues). That is, while participants in the poor condition never get a reliable cue of anyone's accuracy, they do all get multiple poor cues of accuracy and can aggregate these into a potentially reliable prestige cue. If this can be done, high prestige will reflect the collective aggregation of multiple noisy cues of high accuracy and so it is worth attending to.

Lines 181-186: I'd report the Gini coefficients for prestige specific to each of the four conditions. The prediction being that prestige inequality should decline as individuals attend to accuracy information more (thus mitigating Matthew effects from prestige-bias).

We now report the average Gini coefficient for experimental groups across the four conditions and find that it is essentially invariant across conditions (lines 216-220). We further discuss why this is the case, despite participants putting less weight on prestige in the richer conditions (lines 240-245). We conclude that although participants shift away from attending to prestige, they attend instead to accuracy, a factor to which they are even more sensitive (i.e. participants overwhelmingly copy the single most accurate group member), and thus accuracy is capable of driving similar levels of influence inequality, even though Matthew effects do not occur with accuracy.

Lines 240-242: Inequality in who is copied isn't the same as inequality in political decision-making. So I would make this caveat in terms of the relevance of your results to interpretation of hierarchy emergence in the absence of coercion. Also, political deference may be the price to pay for copying someone, but then why should such deference persist beyond the time required to copy the trait in question? It may be that considerable time is required for copying. Considering other determinants of prestige can be helpful here- that individuals defer to others who can be sources of benefit (as a mate, sharing partner, ally, leader) even if those benefits don't come from transferrable skills or knowledge.

We appreciate these concerns and have added nuance throughout our discussion to address them. In particular, we have revised the first paragraph of the discussion to be more careful with the distinction between inequality in prestige and other forms of inequality (lines 299-311). We have also added a sentence discussing the many benefits leaders might bring to their followers beyond accurate information (lines 393-395).

Lines 251-253: Big Men may also be feared, which blurs the lines between dominance and prestige. It is common that Big Men gained influence via a reputation for warriorship, which they then translate into coordination of (competitive) gift-giving later in adulthood, e.g. see <https://pubmed.ncbi.nlm.nih.gov/30868368/>. Also, Big Man leadership is more formalized and competitive relative to leadership that emerges in more egalitarian societies. The label of "Big Man" is generally restricted to those leaders of horticultural or pastoral societies where there is more frequent coalitional violence and more wealth to contest, with greater demand for conflict resolution and collective action. Nevertheless, there is ethnographic evidence that status/influence in more egalitarian settings is also dependent on cooperation, and vice versa cooperation is dependent on the distribution of influence: <https://royalsocietypublishing.org/doi/10.1098/rspb.2019.1367>

We appreciate the reviewer's suggestion and thank them for the papers mentioned. We have reworded this section to make clear that big-men are just one form leaders can take, and that leaders are not solely characterized by generosity and cooperation, but also other traits including warriorship and warfare (lines 315-318). The two papers suggested by the reviewer are now cited in this section.

Lines 257-263/272-276: I recently came across this paper which discusses how the adaptiveness of prestige-bias may depend on presence of other learning mechanisms (https://brill.com/view/journals/jocc/24/5/article-p466_3.xml). It's a puzzle why prestige-bias is adaptive in cases where information quality is poor, as this is when one might expect distribution of prestige to be more decoupled from the distribution of performance, as you found in your experiment! And when information quality is high, why rely on prestige bias at all? Can you address this?

We thank the reviewer for suggesting this paper. Indeed, the limitations it notices in the existing theoretical literature, particularly that prestige is often assumed to correlate with accuracy, are in fact addressed by our final model where the relationship between prestige and accuracy is emergent. We now cite this paper in the discussion (lines 359-361). In addition, we have added to this paragraph to address the reviewer's additional concern; why rely on prestige bias at all? (lines 365-383). To summarize here, we suggest that prestige biased transmission is likely most adaptive for direct cues of intermediate richness, nonetheless there are plausible hypotheses for why prestige should be attended to for a wide range of information richness.

Reviewer #3 (Remarks to the Author):

Review of "Human Prestige Psychology Creates Hierarchical Societies without Coercion.

Overall, this is a very interesting paper and should be published. Using both simulation and experimental data it substantiates a speculation that has existed since the Prestige-Dominance Theory was first developed in the early 2000s.

The biggest challenge the paper confronts is length. In my view, it would be ideal if the authors had more space. Everything feels rather compressed. The paper has multiple simulations and an experiment.

Given space limitations, some of suggestions for material to add might be difficult. But, I'll put it out there anyway.

We thank the reviewer for their positive evaluation of our work and for their helpful comments. We have revised the manuscript to address their concerns as described below, and we hope they find our revisions satisfactory.

On the title: the authors make the case for prestige and individual inequality (Gini). It's important to individual differences in payoffs separate from hierarchy, political power and stratification. Among "big men" societies, the "big man" is in a "first among equals" situation. There's no stratification or hereditary power, etc. Anthropology is full are great examples, but I like the Handbook of California Indians for cases. These population are all foragers, but there are clearly "rich men" and wealthy prestigious foragers.

Anyhow, you should make all this clear. I'd consider sticking to social and economic inequality at the individual level and avoid stratification or hierarchy (which is in the proposed title).

We thank the reviewer for this suggestion (along with a similar suggestion from R1) and have revised the title to "Human Prestige Psychology Creates Adaptive Inequality in Social Influence". This contains the core topic of our work (prestige and social influence), and refers to the distribution of social influence ("Inequality"), as well as our empirical component ("Human") and the final evolutionary model ("Adaptive"). In addition, it avoids the invocation of social stratification or hierarchy. We have also nuanced our language throughout the paper to more accurately reflect the implications of our work.

I also think readers would be helped by learning a little about how prestige is suppressed in foraging societies. Among the Ju'hoansi, the camp "insults" the meat to the hunter, as a way of "cooling his heart" (even if they are delighted). Similarly, credit is shared due to 'arrow sharing' so both the hunter and the owner of the arrow get credit for the kill. This spreads the status bump. Further, hunters who get a couple of kills in a row, take time off so has not to accumulate to much prestige. Boasters are shot-down with jokes if they get too big for their waistcoats. You can see how the norms operate to suppress the prestige that might otherwise result in inequality. This seems important for the argument.

We agree that this is an important point to highlight to readers. We have expanded the discussion (lines 331-345) to include a section describing the practice of the Ju'hoansi as suggested by the reviewer and also a related example observed in the Aché.

I'd also recommend explaining the deference benefits that the prestigious get...additional wives, matings ("wife loaning"), influence at meetings, help cutting canoes (Andaman Islanders). Having many followers improves security, given endemic warfare. This is useful because the simulation and experiments are pretty abstract.

Thank you for this suggestion. We have expanded our description of the benefits of prestige (lines 72-79). Our main example is von Rueden et al's 2011 study of the Tsimané. This is particularly germane because it considered multiple kinds of benefits and also distinguished between the payoffs from prestige and dominance. We additionally provide citations to a number of other examples across different populations.

The main simulation is challenging to explain. I think you can do better. First, create an image of the toroidal grid and show your individuals on the grid. I like Mathematica for this. Some readers were be perplexed by a toroidal grid. In addition to showing it, you should say we you used this. Also, explain why you didn't create communities, like villages, camps or territorial communities (like chimps and bonobos). You seem to spread individuals across space on the donut.

Thank you for these suggestions. We have created an image of a toroidal grid. However, because the mathematics of our simulation don't match many intuitions about a torus (e.g. the inner and outer circumferences are the same length) have paired the 3D torus visualization with another of an "edgeless surface". We have revised the text description to be more agnostic about how to imagine the space while also explaining why we adopted the toroidal ("edgeless") structure (lines 114-119). Given the size of the visualizations, we have placed them in the SI (Fig. S1) and refer to them from the main text (line 117).

We agree that an important robustness check is explore additional spatial configurations of individuals. We have re-run our simulation using the suggested "village" configuration. The results, presented in SI Fig. S5, are qualitatively the same as those presented in the main paper and we refer to them from the main text (lines 146-147).

Two notes: (1) running for 400 generations seems odd. Why not run it until a stable distribution is achieved?

We thank the reviewer for pointing out this omission and now make clear that 400 was chosen precisely because it was more than sufficient for the simulations to reach equilibrium (lines 141-142).

Isn't it crucial to vary the population size? Seems like we want to know how pop size influences things. Some people think repeatedly running a small pop size like 400 is equivalent to running a larger pop size, but its not. Well, sometimes it is, but it can fool you.

We thank the reviewer for this suggestion and have run additional simulations where we varied both the population size and also the size of the space. This allowed us to explore both population size and population density. In all cases the results were qualitatively unchanged. We present these results in SI Fig. S5-S8 and refer to these figures from the main text (lines 146-147).

I wonder if Figure 1 can expanded to deliver a more complete image of the results.

We thank the reviewer for this suggestion. We have moved an additional panel from the supplementary material to the main text which quantifies the extent that the beliefs in the population are spatially structured. Figure 1 now provides a measure of the inequality of the distribution of social influence (panel A), how influence relates to the spatial distribution (panel B) and the spatial structuring of beliefs (panel C). Collectively these provide a comprehensive visualization of the results of model 1.

Experiments—very cool. The experiment confirms the hypothesis model in the simulation. It also nicely confirms the more fundamental prestige-dominance theory.

We thank the reviewer for their appreciation of our experimental results.

Please tell us who these participants are in the main text. I see the info in the supplemental (Mturk and Dallinger), but this is important info. Banishing it to the supplemental implicitly communicates to junior scholars that this represents irrelevant detail.

We appreciate the importance of these details and have added them to the main text (lines 181-184).